# A Genome-Wide Association Study Reveals Region Associated with Seed Protein Content in Cowpea

**DOI:** 10.3390/plants12142705

**Published:** 2023-07-20

**Authors:** Yilin Chen, Haizheng Xiong, Waltram Ravelombola, Gehendra Bhattarai, Casey Barickman, Ibtisam Alatawi, Theresa Makawa Phiri, Kenani Chiwina, Beiquan Mou, Shyam Tallury, Ainong Shi

**Affiliations:** 1Department of Horticulture, University of Arkansas, Fayetteville, AR 72701, USA; yc046@uark.edu (Y.C.); gb005@uark.edu (G.B.); ialatawi@uark.edu (I.A.); tmakawip@uark.edu (T.M.P.); kechiwin@uark.edu (K.C.); 2Texas A&M AgriLife Research, 11708 Highway 70 South, Vernon, TX 76384, USA; waltram.ravelombola@ag.tamu.edu; 3Department of Plant and Soil Sciences, Mississippi State University, North Mississippi Research and Extension Center, Verona, MS 38879, USA; t.c.barickman@msstate.edu; 4USDA-ARS, Crop Improvement and Protection Research Unit, Salinas, CA 93905, USA; beiquan.mou@usda.gov; 5USDA-ARS, Plant Genetic Resources Conservation Unit, 1109 Experiment Street, Griffin, GA 30223, USA; shyam.tallury@usda.gov

**Keywords:** GWAS, cowpea, seed protein content, genomic prediction

## Abstract

Cowpea (*Vigna unguiculata* L. Walp., 2*n* = 2*x* = 22) is a protein-rich crop that complements staple cereals for humans and serves as fodder for livestock. It is widely grown in Africa and other developing countries as the primary source of protein in the diet; therefore, it is necessary to identify the protein-related loci to improve cowpea breeding. In the current study, we conducted a genome-wide association study (GWAS) on 161 cowpea accessions (151 USDA germplasm plus 10 Arkansas breeding lines) with a wide range of seed protein contents (21.8~28.9%) with 110,155 high-quality whole-genome single-nucleotide polymorphisms (SNPs) to identify markers associated with protein content, then performed genomic prediction (GP) for future breeding. A total of seven significant SNP markers were identified using five GWAS models (single-marker regression (SMR), the general linear model (GLM), Mixed Linear Model (MLM), Fixed and Random Model Circulating Probability Unification (FarmCPU), and Bayesian-information and Linkage-disequilibrium Iteratively Nested Keyway (BLINK), which are located at the same locus on chromosome 8 for seed protein content. This locus was associated with the gene *Vigun08g039200*, which was annotated as the protein of the thioredoxin superfamily, playing a critical function for protein content increase and nutritional quality improvement. In this study, a genomic prediction (GP) approach was employed to assess the accuracy of predicting seed protein content in cowpea. The GP was conducted using cross-prediction with five models, namely ridge regression best linear unbiased prediction (rrBLUP), Bayesian ridge regression (BRR), Bayesian A (BA), Bayesian B (BB), and Bayesian least absolute shrinkage and selection operator (BL), applied to seven random whole genome marker sets with different densities (10 k, 5 k, 2 k, 1 k, 500, 200, and 7), as well as significant markers identified through GWAS. The accuracies of the GP varied between 42.9% and 52.1% across the seven SNPs considered, depending on the model used. These findings not only have the potential to expedite the breeding cycle through early prediction of individual performance prior to phenotyping, but also offer practical implications for cowpea breeding programs striving to enhance seed protein content and nutritional quality.

## 1. Introduction

Cowpea (*Vigna unguiculata* L. Walp., diploid, 2*n* = 2*x* = 22) is a protein-rich crop that complements staple cereals for humans and is fodder for livestock [1,2]. In the semi-arid regions of sub-Saharan Africa, cowpea can still reach an output of 1000 kg/ha, which provides millions of people with cheap, high-quality food protein [3]. The protein content of cowpea is between 20.3 and 32.5% (on a dry-weight basis), higher than that of many other legumes such as chickpea (*Cicer arietinum*), 13.3–26.8% [4], and lima bean (*Phaseolus lunatus*), 20.7–23.1% [5]. The high-quality plant-based protein contains all nine essential amino acids required for human health. Adding cowpea to children’s diets can significantly reduce developmental diseases such as malnutrition and infection in developing areas [6]. Its protein can also be used in various food products, including meat alternatives [7], baked goods [8], and protein bars [9], creating economic opportunities for food companies and entrepreneurs [10]. Assessing the protein content of cowpea germplasm can aid plant breeders in identifying and developing cultivars with high seed-protein content within their breeding programs. Ddamulira et al. [11] measured the protein content of 30 cowpea genotypes and found that the average cowpea protein content was between 23.9% and 30.3%. Weng et al. [12] developed a Near-Infrared Reflectance (NIR) rapid method to analyze 240 cowpea genotypes and found the protein contents were 28.8–37.8%. Boukar et al. [13] measured the protein content of 1541 germplasm lines, and the results were 17.5–32.5%, with an average of 25.0% and a standard deviation of 0.4 g.

Breeding efforts for cowpea protein content traits have been ongoing since the last century. B.B. Singh et al. [14] improved 52 cowpea varieties and obtained three varieties, IT89KD-245, IT89KD-288, and IT97K-499-35, with a protein content of 26%. Raina et al. [15] used sodium azide and radiation-induced mutation to improve cowpea protein content. However, the development of a new cowpea cultivar through breeding can take many years, typically around 8 to 10 years, or even longer. Therefore, MAS is a good approach for plant breeders that can accelerate the development of new and improved plant varieties with desirable traits [16].

Linkage- and family-based genetic mapping have proven effective in detecting quantitative trait loci (QTLs) that have major or minor phenotypic influences for simple and complex traits in cowpea. Kongjaimun et al. [17] utilized 226 SSR markers from related *Vigna* species and identified one major and six minor QTLs for the variance in pod length between yard long bean and wild cowpea. Andargie et al. [18] employed SSR markers to identify six QTLs for seed size and four QTLs for pod shattering, with phenotypic variation ranging from 8.9% to 19.1% and 6.4% to 17.2%, respectively. Using RFLP markers, Fatokun et al. [19] established genomic maps for cowpea and detected major QTLs for seed weight. Furthermore, Muchero et al. [20] mapped 12 QTLs associated with seedling drought tolerance and maturity in a cowpea recombinant inbred (RIL) population.

Genome-wide association studies (GWASs) are a powerful tool used in genetics research to identify genetic variations associated with various traits of interest [21]. Several GWASs in cowpea have been conducted using different methods and different populations. Huynh et al. [22] used 51,128 single-nucleotide polymorphisms on 368 diverse cowpea accessions from 51 countries to identify 17 loci related to seed weight, length, and width. Burridge et al. [23] identified 11 significant loci associated with biologically relevant variation in cowpea root architecture with a 189-entry cowpea diversity panel. Paudel et al. [24] performed a GWAS to identify marker trait associations for flowering time in 292 cowpea accessions using 51,128 SNPs, resulting in the identification of 7 reliable SNPs and candidate genes. Wu et al. [25] combined high-throughput physiological phenotyping of 106 cowpea accessions under progressive drought stress with a GWAS, which allowed for the genetic mapping of complex drought-responsive stomatal traits and the identification of a final set of 30 significant SNPs associated with stomatal closure, providing a new methodology for exploring the genetic determinants of water budgeting in crops under stressful conditions. Kpoviessi et al. [26] evaluated 107 cowpea collections from six countries for their responses to *Callosobruchus maculatus* to identify three quantitative trait nucleotides (QTNs) which were linked to candidate genes located nearby. However, compared with context traits or protein contents in other leguminous crops, the research on cowpea protein-content breeding is obviously insufficient since no GWASs have been reported on this aspect.

Genomic selection (GS) is a relatively new approach in plant breeding that allows for the selection of desirable traits based on genomic information and has the potential to accelerate the development of new and improved cowpea varieties with improved resistance to pests and diseases, drought tolerance, and other desirable traits. However, very few studies were reported on the applications of GS or GP in cowpea, such as a study to evaluate considerations for genetic architecture in GS models for flowering time, maturity, and seed size and a study to conduct GS for the drought tolerance indices [27]. Additionally, there have been no studies reported on cowpea seed quality so far.

The objectives of this study were to conduct a GWAS to identify SNP markers associated with the seed protein content of cowpea and to estimate the GS accuracy of predicting protein content using a 161-cowpea population as the first report.

## 2. Results

### 2.1. Phenotypes

The nitrogen contents were assessed from three replicates and two locations, following a randomized complete block design for the implementation of the field experiment. Subsequently, the protein content data for each accession of cowpeas was estimated and obtained (Appendix A). The distribution and accumulation density of the protein contents are shown in Figure 1. In total, 24 accessions were under 24.00%, 35 were from 24.00 to 25.00%, 30 were from 25.00 to 26.00%, 44 were from 26.00 to 27.00%, and 28 were more than 28.00%. Among the 161 accessions, the highest protein content was PI662992, reaching 28.87%. The lowest was PI339587, and the protein content was 21.80%. The average protein content was 25.61%, and the standard deviation was 1.49%. The protein content of the 161 accessions followed an approximately normal distribution. The estimated broad-sense heritability (h^2^) was found to be 53.8%, indicating a moderately high level of inheritability in seed protein content of cowpea.

### 2.2. SNP Profile

For the GWAS and GP in this study, a collection of 110,155 high-quality single-nucleotide polymorphisms (SNPs) were employed, and their distribution across the 11 chromosomes is depicted in Figure 2. The average inter-SNP distance ranged from 3.3 kb to 6.5 kb across each chromosome, with an overall mean of 3.9 kb. The average minor allele frequency (MAF) across the entire genome was 21.6%, while the rates of heterozygosity and missingness were 2.4% and 0.3%, respectively.

### 2.3. Population Structure Analysis

The population structure of the 161 cowpea accessions was initially inferred using STRUCTURE 2.3.1 and the peak of delta K was observed at K = 2, by 110,155 high-quality SNPs indicating the presence of two sub-populations. At a threshold value of 0.5, 79 of the 161 accessions (49.1%) were assigned to Q1 subpopulation; 82 accessions (50.9%) were assigned to Q2 (Appendix A). Phylogenetic analysis and a population admixture map of the 161 accessions using the GAPIT 3 R package also showed a clustering pattern consistent with that inferred by structure K = 2 (Figure 3A). The two groups were also observed based on PCA dimensions (Figure 3B). The most closely related accessions based on Structure analysis were grouped in the neighbor branches of the phylogenetic tree using Neighbor-Joining analysis (Figure 3C). Therefore, the 161 accessions can be divided into two sub-populations based on both structural and phylogenetic analyses. The kinship matrix, based on 110,155 SNPs for the studied genotypes, indicated that there was no clear clustering among the 161 genotypes (Appendix A).

### 2.4. GWAS Analysis and Candidate Gene

Association analysis for cowpea protein content was conducted by using 110,155 high-quality SNPs using four methods of MLM, GLM, Blink, and FarmCPU of GAPIT3 and three methods of SMR, GLM, and MLM of TASSEL5.0. In this study, the QQ plots (Figure 4A and Appendix A) demonstrated a significant deviation from the expected distribution of the observed *p*-value, suggesting the presence of SNPs was associated with protein contents in this population. The results of GAPIT3 (Figure 4C) and TASSEL5.0 (Appendix A) showed a high consistency of significant SNPs with protein contents which were located in chromosome 8. In total, seven SNPs with higher LOD and MAF values were identified: Vu08_3838280, Vu08_3838282, Vu08_3838296, Vu08_3839577, Vu08_3839579, Vu08_3840180, and Vu08_3840193 (Table 1), which were located between the 3839 kb and 3841 kb physical position of chromosome 8 (Figure 4B). Moreover, based on the analysis of haplotype blocks, the seven SNPs were located at the same block, which indicated that the SNPs tend to be inherited together, rather than being shuffled independently during meiosis. We identified Vu08_3839577 as the most significant SNP in comparison to the others. Specifically, the FarmCPU and Blink models of GAPIT3 produced LOD scores of 10.78 and 6.60, respectively, in association with this SNP. Moreover, the R^2^ of Vu08_3839577 was up to 22.44%, with similar values in the other six loci.

Given that all significant SNPs were located within a single block (Figure 4B), candidate genes were identified within the region of the block, encompassing a physical distance of 30 kb on either side. A total of three candidate genes, *Vigun08g039100*, *Vigun08g039200*, and *Vigun08g039300*, were identified and located at the positions of −25 kb, −6 kb, and 6 kb beside the haplotype block (Table 2). The candidate genes were annotated as Fructan fructosyltransferase (*Vigun08g039100*), hioredoxin superfamily protein, glutaredoxin subgroup III (*Vigun08g039200*), and heat shock transcription factor A2 (*Vigun08g039300*), respectively.

### 2.5. Genomic Prediction Analysis

In this study, eight SNP sets were used for GP analysis using five models: BA, BB, BL, BRR, and rrBLUP (Figure 5). The average prediction accuracies of random SNP sets ranged from 18.4% to 53.2%, and generally increased with the number of SNPs included, increasing from 7 to 10 k. The similar average accuracies among all models by randomly SNP sets ranged from 39.1% (rrBLUP) to 45.1% (BL). The set of seven SNPs associated with a trait had accuracies comparable to the larger sets of random SNPs and was particularly strong in the BL and BRR models (Appendix A). Therefore, using trait-associated marker alleles to perform GP is more efficient for selecting protein content in cowpea breeding.

## 3. Discussion

### 3.1. Population and Phenotyping

Cowpea is a significant food crop in tropical and subtropical regions, but research on seed protein content in this crop remains limited compared with other legume species such as soybean [28,29]. In the present investigation, a set of 161 cowpea germplasm accessions obtained from 31 countries was analyzed, manifesting substantial genetic diversity and encompassing 10 distinct seed coat types, as previously documented by Xiong et al. [30,31]. This diverse genetic reservoir offers a promising avenue for a comprehensive exploration of the genetic determinants underlying seed protein content, consequently emphasizing the noteworthy variability inherent within the crop. The scarcity of data on germplasms with high seed-protein content available for cowpea breeding programs makes it crucial to screen cowpea germplasms to identify elite genotype(s) with high protein contents [32]. The protein content of cowpea seeds is an essential index that is closely related to quality, health, nutrition, and market price, regardless of whether the seeds are used for direct human consumption or processed into flour for baked goods or other products. Farmers and consumers prefer high-protein varieties and products; thus, achieving a high protein content has become a crucial goal in cowpea breeding programs and production [33]. According to previous research, cowpea seeds typically contain 25% protein [13]. However, identifying varieties and germplasms with higher protein contents than the average of 25% could be beneficial [34]. Asante et al. conducted a study on 32 cowpea accessions in Ghana to investigate the variation in protein content and found that the seed protein content ranged from 16.4% to 27.3%, with an average of 22.5% [35]. A total of 28 elite USDA cowpea accessions exhibiting a seed-protein content exceeding 28% were identified, surpassing that of conventional commercial cowpea cultivars. These accessions could prove valuable for utilization in Marker-Assisted Selection (MAS) breeding to develop novel cultivars with elevated protein content.

Cowpea seed protein content has been found to be highly heritable. Ajeigbe et al. [36] reported a broad-sense heritability of 86% based on nine cowpea varieties. Similarly, Nielsen et al. [37] observed a high broad-sense heritability of 95% for seed protein in 100 cowpea lines based on data from a single location. Tchiagam et al. [38] determined a broad-sense heritability of 74% for seed protein using five divergent lines for cross mating. Emebiri [39] reported broad-sense heritability for protein content ranging from 70% to 78% in two crosses. Moreover, our study found a lower heritability estimate of 53.8% for cowpea seed protein content compared with previous reports (>75%). This is attributed to the two-location trials in our analysis. If we independently calculated the data collected from each location, the heritability estimate would have exceeded 80%.

### 3.2. The Models of GWASs

GWASs have emerged as a powerful tool for identifying genetic variants associated with complex traits, including protein content in leguminous crops [40]. Due to the complex nature of genetic architecture and environmental factors, it is not uncommon for different statistical models to yield slightly different results. The use of multiple statistical models in GWASs is common practice, as it helps to reduce the risk of false-positive associations and increases the robustness of the results [41,42,43]. However, the fact that all models produced similar results in a study provides additional confidence in the validity of the findings [44]. In this report, the GWAS identified several SNPs associated with protein content using multiple models, and interestingly, all models yielded similar results. This finding suggests that the association between the genetic variants and the trait is robust and not dependent on the choice of statistical model.

### 3.3. The SNPs Associated with the Seed Protein Content

Several studies have applied GWASs to investigate the genetic basis of protein content in various leguminous crops. Priyanatha et al. [45] utilized a genomic panel consisting of 200 genotypes to investigate yield, protein, and oil concentrations using the FarmCPU model of GWAS. Hwang et al. [46] performed a GWAS to identify quantitative trait loci (QTL) controlling seed protein and oil concentration in 298 soybean germplasm accessions. Zhang et al. [47] identified three QTLs related to protein content using a GWAS with 211 diverse soybean accessions genotyped with a 355 K SoySNP array. Lee et al. [48] conducted a GWAS using phenotypic data collected from five environments for 621 accessions and 34,014 markers to identify three QTLs for seed protein content. Upadhyaya et al. [49] identified seven genomic loci associated with seed protein content using 16,376 genome-based SNPs in 336 sequenced chickpea accessions. However, there is currently no report on cowpea protein content using a GWAS. In this study, the analysis of cowpea seed protein content in 161 accessions under multi-models is the first report of a GWAS in this field.

In total, seven SNPs were identified that were located in the same haplotype block. This finding can provide valuable insights into genetic architecture [50]. It may suggest that these SNPs are tagging a common causal variant located within this block, or that the block itself contains functional elements that affect the trait [51]. Further investigations are necessary to identify the specific variant(s) responsible for the observed association, but the identification of these SNPs in the same haplotype block provides an important starting point for this process.

### 3.4. The Candidate Genes

Three candidate genes were identified by seven significance SNPs and were annotated as Fructan fructosyl-transferase (*FFT Vigun08g039100*), thioredoxin superfamily protein/Glutaredoxin subgroup (*Vigun08g039200*), and heat shock transcription factor A2 (*HSFA2 Vigun08g039300*).

The Thioredoxin superfamily is an essential group of proteins that regulate seed protein content in plants [52]. Seed protein content is regulated by the balance between protein synthesis and degradation, which is controlled by various factors, including the Thioredoxin superfamily. Studies have highlighted the importance of Thioredoxin h (Trx h) and the Glutaredoxin subgroup (GrxS) in regulating seed protein content [53]. Trx h plays a significant role in regulating the accumulation of storage proteins in seeds by controlling the expression of genes involved in the synthesis and accumulation of these proteins [54]. GrxS, on the other hand, regulates the degradation of storage proteins by controlling the activity of cysteine proteases that break down proteins [55,56]. Other members of the Thioredoxin superfamily, such as Thioredoxin m and Thioredoxin x, also play a role in regulating seed protein content [57]. In summary, the Thioredoxin superfamily has diverse functions in regulating seed protein content, making it essential for seed development and quality.

FFT is an enzyme involved in the biosynthesis of fructans, which are carbohydrate molecules found in plants [58]. HSFA2 is a transcription factor that plays a crucial role in the response of plants to heat stress [59]. There is no direct relationship between FFT/HSFA2 and protein content in plants. It is worth noting that there may be indirect effects on protein content in plants. For example, fructans can affect plant growth and development, which in turn can impact the expression of genes involved in protein synthesis and degradation pathways [60]. Additionally, fructans may provide a source of energy for plant metabolism, which could indirectly support protein synthesis [61]. Additionally, studies have shown that overexpression of HSFA2 in Arabidopsis leads to an increase in the accumulation of several classes of proteins [62]. Additionally, the overexpression of HSFA2 in rice resulted in an increase in the accumulation of storage proteins in the seeds [63]. Nonetheless, the relationship between FFT/HSFA2 and protein content in plants is complex and requires further investigation.

### 3.5. The Genomic Prediction

The study identified seven significant SNPs located within a single locus containing genes that are associated with storage proteins. However, prior to the application of these findings in breeding, further verification work is required [64,65]. In recent years, GS has gained popularity in large-scale crop-breeding programs. Previous studies have indicated that GS achieves a more robust prediction of genotypic values when compared with QTLs for traits controlled by numerous genes with small effects. GS is considered to offer a superior and more reliable prediction of outcomes than the traditional QTL approach, as it employs more markers that are distributed throughout the genome and captures more of the genetic variation of a trait [66]. Furthermore, GS can enable predictions of an individual’s performance prior to phenotyping, potentially saving up to 50% of the time and resources required in the breeding process [67].

However, there is currently no existing research investigating the effectiveness of GS or GP for cowpea seed protein content. To address this gap, we conducted a study utilizing GP with five models based on GWAS-derived SNPs and seven randomly selected sets containing between 7 and 10,000 SNPs. The accuracy of the GWAS-based SNPs ranged from 42.9% to 52.1%, which is similar to the accuracies reported in previous studies on seed protein contents of other plant species like winter wheat [68] and soybean [69,70], but lower than the prediction of flax [71]. Notably, the accuracy of the GP based on the GWAS-derived SNPs was higher than that of randomly selected SNP sets containing 7, 200, and 500 SNPs, and was comparable to that of random SNP sets containing ≥1000 SNPs. These findings suggest that significant SNPs derived from GWAS are important and efficient for use in breeding selection of seed protein content. The candidate SNPs are centered on chromosome 8, located between bases 3,839,000 and 3,841,000. This indicates that there may be a QTL regulating cowpea protein around these two kilobases.

In summary, the tight clustering of seven SNP markers within a small genomic region, with Vu3839577 as the peak marker, indicates a major QTL governing cowpea protein content. By focusing on this marker and exploring its effects and potential use, breeders can streamline selection processes and make significant strides towards developing high-protein cowpea varieties with broader agricultural and nutritional benefits. Further investigations into its biological mechanisms and interactions could enhance our understanding of cowpea genetics and support targeted crop-improvement efforts.

## 4. Materials and Methods

### 4.1. Plant Materials and Field Experiment

In this study, 161 cowpea genotypes including 151 USDA germplasm collections and 10 Arkansas lines were assessed. The cowpea plants were cultivated using a randomized complete block design (RCBD) with three replications in a single 14-foot-long row with a 3 foot row spacing and approximately 4 inches plant spacing at two distinct locations in Arkansas: Fayetteville (36°4′ N, 94°9′ S) and Alma (35°29′ N, 94°13′ S) in 2016. Throughout the growing season, no pesticides, herbicides, or chemicals were employed to manage pests, diseases, or weeds. No regular irrigation was upheld before maturity. Harvesting was carried out by bulk harvesting cowpea pods when 90% of pods dried at maturity stages. The cowpea seeds were subsequently shelled and cleaned following the harvest of pods [72].

### 4.2. Seed Protein-Content Assessment

A total of 966 samples were collected from 161 cowpea accessions with three replications at two locations, as described above. In order to measure seed protein content, each cowpea genotype sample was carefully selected based on matured seeds, uniform color and size, and the absence of damage from insects or machinery. Approximately 20 g of cowpea seeds from each sample was ground using a coffee grinder (Hamilton Beach, MODEL: 80335RV, Glen Allen, VA, USA) for 1 min. Next, 5 g of the ground powder was sieved through a 100# sieve (nominal wire diameter 0.1 mm), and each sample was weighed as 1 g, then transferred to a 0.2 mL microfuge tube for protein determination. The cowpea seed protein content was determined through the analysis of nitrogen percentage by combustion using an Elementar Rapid N III instrument (Elementar, Rhine Main, Germany) at the Agriculture Diagnostic Laboratory, University of Arkansas. We loaded each representative powdered sample into the instrument’s combustion chamber and allowed the instrument to analyze the sample by measuring released nitrogen gas. Combustion was performed at high temperature and in the presence of pure oxygen to remove nitrogen, which was subsequently isolated from other combustion products. The nitrogen content was then measured with a thermal conductivity detector for each sample, and the percentage of nitrogen in each sample was determined [73]. Finally, the total protein content for each sample was estimated by multiplying the nitrogen content with the conversion factor as 6.25 [74]. The phenotypic data were analyzed with SAS 9.2 (SAS Institute, Cary, NC, USA) software. The formula of heritability (*h*^2^) was used to determine each trait.
(1)$$h^{2}=\frac{\sigma_{G}^{2}}{\sigma_{G}^{2}+\left(\frac{\sigma_{GE}^{2}}{e}\right)+\left(\frac{\sigma_{E}^{2}}{re}\right)}$$
where $\sigma_{G}^{2}$ is the genetic variance; $\sigma_{GE}^{2}$ is genotype × environment variance; $\sigma_{\varepsilon }^{2}$ is the residual variance; *e* is the number of environments (locations); and *r* is the number of replications (blocks).

### 4.3. DNA Extraction and Construction of Gene Library and GBS

Fresh leaf genomic DNA was extracted from freeze-dried young cowpea leaves using the CTAB (cetyltrimethylammonium bromide) protocol [75]. DNA content was detected using a NanoDrop 200 c spectrophotometer (Thermo SCIENTIFIC, Wilmington, DE, USA). The DNA library was obtained by treating the DNA with ApeKI restriction endonuclease. DNA normalization, library preparation, and GBS (genotyping by sequencing) were conducted with HiSeq 2000 in the Beijing Genome Institute (BGI), China. The cowpea reference genome was provided by Dr. Timothy J. Close, University of California, Riverside [76]. After screening SNP data with minor allele frequency (MAF) > 5% and missing data < 10%, 110,155 high-quality SNPs were finally obtained.

### 4.4. Population Structure and Genetic Diversity

LEA is an R package designed for conducting population structure and genomic signature analysis of local adaptations. The inference algorithms utilized by R are based on a fast version of the structure algorithm, which is available through the LEA package [77]. The Structure analysis identifies K clusters by measuring an optimum ΔK based on the SNP data provided. A preliminary analysis was performed in multiple runs by inputting successive values of K from 2 to 20. Once an optimum K was determined, each cowpea accession was assigned to a cluster (Q) based on the probability that the accession belonged to that cluster, with a cut-off probability for the assignment set to 0.5. Using the optimum K, a bar plot with ‘Sort by Q’ was generated to visualize the population structure among the 161 accessions. Additionally, phylogenetic relationships and principal component analyses (PCAs) among the accessions were generated and drawn using the R package GAPIT 3 (Genomic Association and Prediction Integrated Tool version 3, https://zzlab.net/GAPIT/index.html; https://github.com/jiabowang/GAPIT3, accessed on 1 April 2023) [78]. During the drawing of the phylogeny trees and PCA, the population structure and cluster information were imported for the combined analysis of genetic diversity. For the sub-tree of each Q (cluster), the shape of ‘Node/Subtree Marker’ and the ‘Branch Line’ was drawn using the same color scheme as the STRUCTURE analysis.

### 4.5. GWAS and Candidate Gene

The association analysis of the cowpea dataset was conducted using TASSEL 5.0 software [79]. Three different association analysis models were employed, including single-marker regression (SMR), the General Linear Model (GLM), and Mixed Linear Model (MLM) [80]. Additionally, two other models, Fixed and Random Model Circulating Probability Unification (FarmCPU) [81] and Bayesian-information and Linkage-disequilibrium Iteratively Nested Keyway (BLINK) [78], were utilized in R software GAPIT 3 [78]. The models were set with principal component analysis (PCA) equal to two, and pseudo QTNs were employed. Haplotype blocks (HAP) were estimated using Plink 2.0 software [82] within 100 kb, and a minimum threshold value of 0.05 for minor allele frequency (MAF) was used. Candidate genes were selected based on the peak significant SNP in each linkage disequilibrium (LD) region located within 30 kb on either side of significant SNPs [67]. The candidate genes were retrieved from the reference annotation of the cowpea reference genome Vigna unguiculata v1.2 from Phytozome database (https://phytozome.jgi.doe.gov).

### 4.6. Genomic Prediction

GP was conducted using eight genotype datasets, including seven randomly selected SNP sets (7, 200, 500, 1 k, 2 k, 5 k, and 10 k SNPs) and a trait-associated marker set (seven SNPs) according to the GWAS results. Genomic estimated breeding value (GEBV) was computed using five different statistical models, namely, ridge regression best linear unbiased predictor (rrBLUP) [83], Bayes ridge regression (BRR), ‘Bayes A’ (BA), ‘Bayes B (BB)’ Bayesian least absolute shrinkage, and selection operator (BL) [84]. A five-fold cross validation to a training/testing set as 20%/80% was performed for the genomic prediction study. The association panel was randomly divided into five disjointed groups. A total of 100 replications were conducted at each fold. Mean and standard errors corresponding to each fold were computed [85].

## 5. Conclusions

This study utilized GWAS to identify seven significant SNP markers located at a single locus on chromosome 8 associated with seed protein content. Further analysis revealed that the gene *Vigun08g039200*, annotated as a protein of the thioredoxin superfamily, plays a critical role in improving seed protein content and nutritional quality. To assess the accuracy of predicting seed protein content in cowpea, a GP approach was employed. The GP results showed that the accuracies of predicting seed protein content varied between 42.9% and 52.1%, depending on the model used. The findings suggest that GP is a useful tool for breeders to predict the selection accuracy of complex traits such as seed protein content in cowpea. Moreover, this approach may help expedite the breeding cycle by enabling early prediction of individual performance before phenotyping. Overall, these results provide practical implications for cowpea breeding programs seeking to enhance seed protein content and nutritional quality.

## Figures and Tables

**Figure 1 plants-12-02705-f001:**
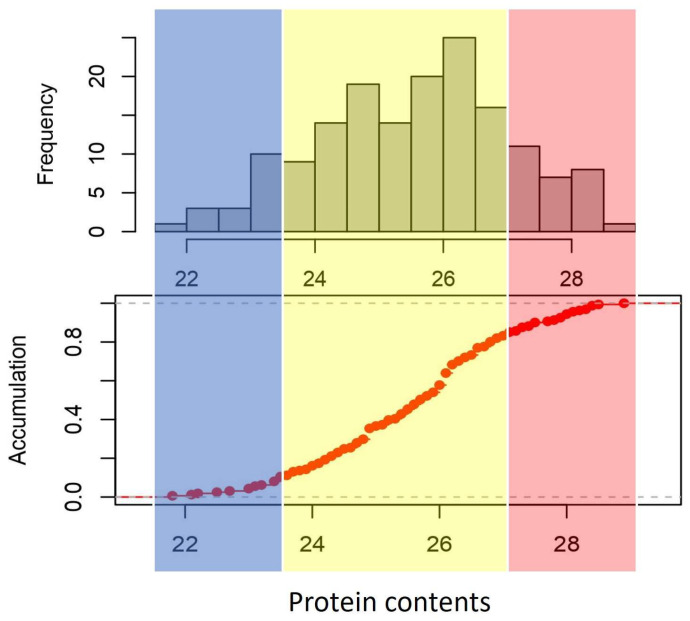
The distribution and accumulation density of the protein contents in 161 cowpea accessions. Blue: low protein contents, yellow: medium protein contents, red: high protein contents.

**Figure 2 plants-12-02705-f002:**
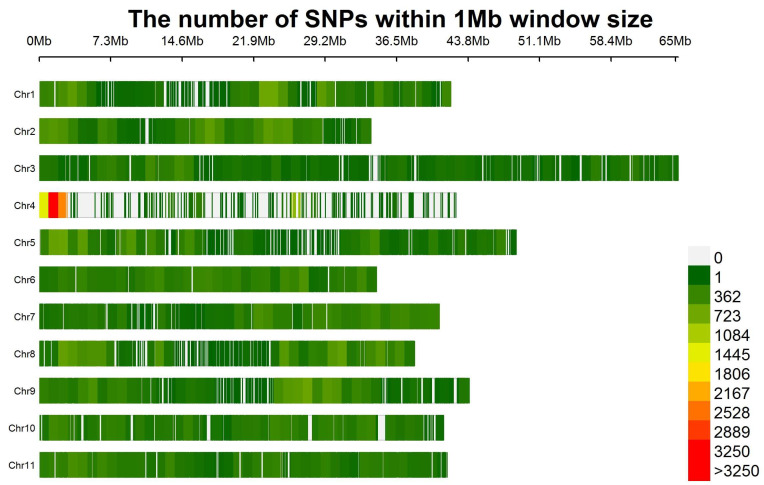
The distribution of 110,155 SNPs among the 11 chromosomes of cowpea within 1 Mb size.

**Figure 3 plants-12-02705-f003:**
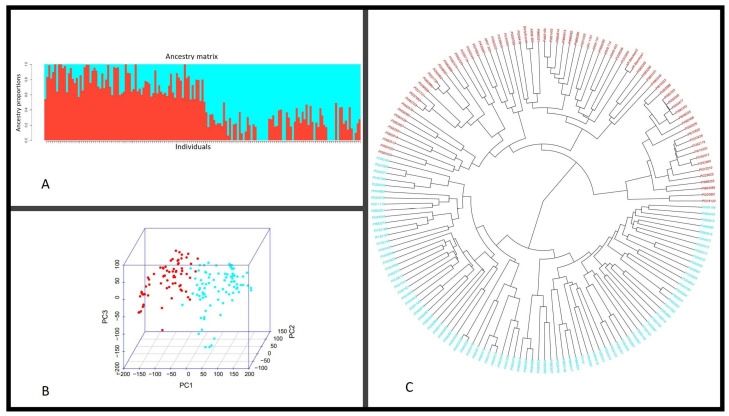
The structure, principal component, and phylogenetic analysis of 161 cowpea accessions were based on 110,155 SNPs. (**A**) Classification of 161 accessions in two groups (K = 2) using STRUCTURE. The distribution of accessions to different populations is color−coded. The *X*−axis represents the 161 accessions, and the value on the *Y*−axis shows the likelihood of every individual belonging to one of the two colored subpopulations, Q1 = red, Q2 = cyan; (**B**) scatter diagram of PCA for 161 accessions labeled by Q groups with the colors in (**A**); (**C**) phylogenetic analysis of the 161 with the corresponding labels as Q group colors in (**A**).

**Figure 4 plants-12-02705-f004:**
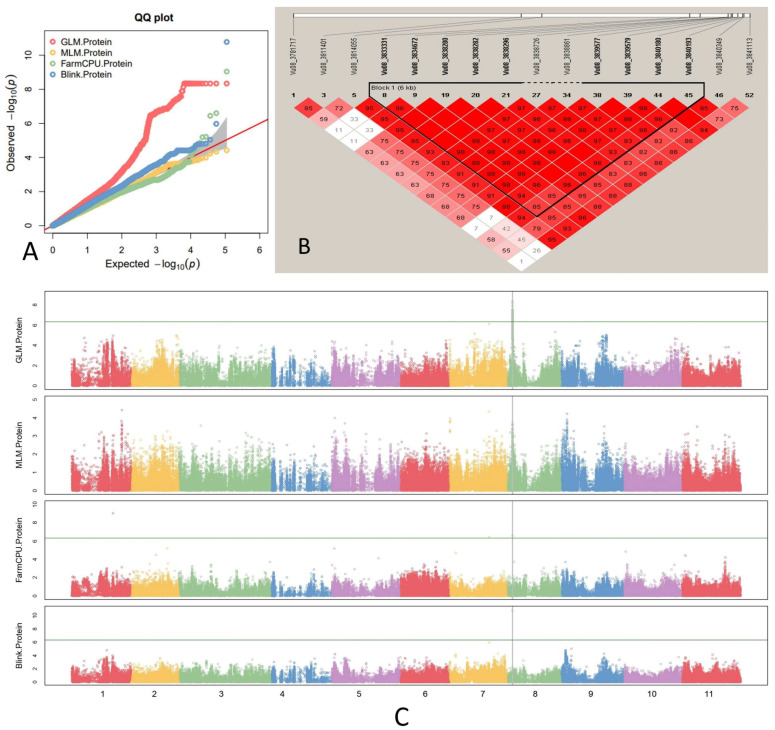
The QQ plot (**A**), haplotype block (**B**), and Manhattan plots (**C**) for cowpea protein contents using four GWAS models: Bayesian-information and Linkage-disequilibrium Iteratively Nested Keyway (BLINK), Fixed and Random Model Circulating Probability Unification (FarmCPU), the Mixed Linear Model (MLM), and the Generalized Linear Model (GLM) by GAPIT 3.

**Figure 5 plants-12-02705-f005:**
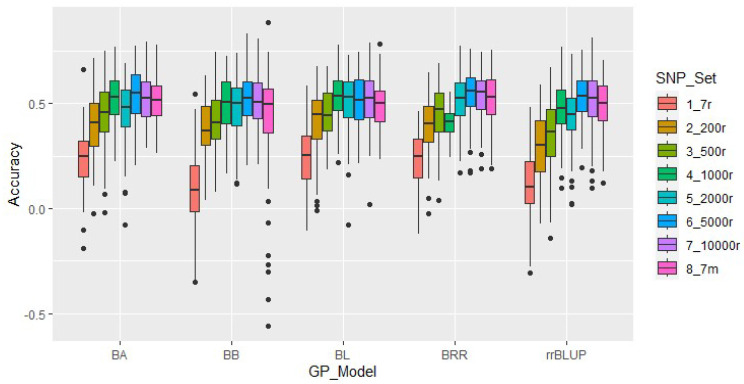
Genomic prediction (GP) accuracy (r−value) for protein contents using five GP models, ridge regression best linear unbiased predictor (rrBLUP), Bayes ridge regression (BRR), ’Bayes A’ (BA), ‘Bayes B‘ (BB), and Bayesian least absolute shrinkage and selection operator (BL), on GWAS−based markers (7 m = violet) and randomly selected SNP sets (7 r = red, 200 r = tan, 500 r = kelly, 1000 r = green, 2000 r = teal, 5000 r = blue, and 10,000 r = purple).

**Table 1 plants-12-02705-t001:** SNP markers associated with seed protein content in cowpea, based on four models, BLINK, FarmCPU, MLM, and GLM, in GAPIT 3 and three models, MLM, GLM, and SMR, in Tassel 5, and *t*-test.

SNP	Chr	Position	−Log(*p*-Value) Using GAPIT 3				−Log(*p*-Value) in Tassel			*t*-Test	Rsq in Tassel			High Protein Content Allele	Low Protein Content Allele	MAF (%)
			Blink	FarmCPU	MLM	GLM	SMR	GLM	MLM	−LOG(*p*)	SMR	GLM	MLM			
Vu08_3838280	8	3,838,280	0.04	0.28	3.22	7.89	10.59	9.19	3.35	13.94	26.56	21.56	10.21	T	A	47.52
Vu08_3838282	8	3,838,282	0.04	0.28	3.22	7.89	10.59	9.19	3.35	13.94	26.56	21.56	10.21	T	A	47.52
Vu08_3838296	8	3,838,296	0.16	0.19	3.33	8.14	10.92	9.54	3.42	14.28	27.26	22.26	10.44	C	G	46.89
Vu08_3839577	8	3,839,577	10.78	6.60	3.62	8.34	10.95	9.63	3.20	14.50	27.33	22.44	9.755	G	A	46.58
Vu08_3839579	8	3,839,579	0.00	0.00	3.62	8.34	10.95	9.63	3.20	14.50	27.33	22.44	9.755	T	C	46.58
Vu08_3840180	8	3,840,180	0.35	0.25	3.59	6.55	9.04	6.95	3.01	12.64	23.17	16.82	9.144	A	G	44.10
Vu08_3840193	8	3,840,193	0.48	0.15	3.75	7.14	9.68	7.98	3.60	13.09	24.59	19.03	11.04	C	A	49.69

**Table 2 plants-12-02705-t002:** Functional annotation of the genes within the 50 kb genomic region harboring the significant SNPs that associate with seed protein content.

Gene	Function	Chr	Gene Start Pos	Gene End Pos	SNP	Chr	Pos	Distance (Bp) from Gene Start and End	
*Vigun08g038900*	Fructan fructosyltransferase	Vu08	3,789,846	3,793,119	Vu08_3839577	8	3839577	−49,731	−46,458
*Vigun08g039000*	Fructan fructosyltransferase	Vu08	3,799,649	3,802,436				−39,928	−37,141
*Vigun08g039100*	Fructan fructosyltransferase	Vu08	3,814,161	3,817,053				−25,416	−22,524
*Vigun08g039200*	Thioredoxin superfamily protein, OsGrx_C15—Glutaredoxin subgroup III, expressed	Vu08	3,832,765	3,834,779				−6812	−4798
*Vigun08g039300*	Heat shock transcription factor A2	Vu08	3,846,346	3,848,728				6769	9151
*Vigun08g039400*	Thromboxane-A synthase/Thromboxane synthetase	Vu08	3,869,695	3,873,523				30,118	33,946
*Vigun08g039500*	IQ calmodulin-binding motif domain containing protein, expressed	Vu08	3,873,436	3,876,766				33,859	37,189
*Vigun08g039600*	LOC_Os06g05730, expressed protein	Vu08	3,887,821	3,888,663				48,244	49,086

## Data Availability

The data that support the findings of this study are available in the Appendix A.

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
