# Peer review of "A Genome-Wide Association Study Reveals Region Associated with Seed Protein Content in Cowpea"

_plants, 2023, doi:10.3390/plants12142705_

Round 1

Reviewer 1 Report

This manuscript describes a GWAS and genomic prediction of protein content in cowpea. In the study, a population of 161 germplasm accessions were genotyped with a total of 110,155 polymorphic SNPs and phenotyped in a replicated field trial at two locations. Different statistical models and methods were used to validate the results. Seven significant SNP markers associated with protein content were detected and seven candidate genes were identified as well. It also indicated that genomic selection or prediction would be effective and applicable. The study will be helpful for cowpea genetic research and breeding for protein improvement. Therefore, this manuscript can be accepted for publication in the journal Plants after some revisions have been made. I have following points for the authors to consider in revising the manuscript:

1.      The population size seems a little bit small, consisting of 161 germplasm accessions, especially compared with other major crops like corn and wheat. However, it should be OK as cowpea is a minor crop and has not been investigated so extensively, in particular its germplasm pool is limited. The authors may discuss more about the diversity and the genetic variability of this population.

2.      Because all the seven SNP markers identified to be associated with the trait (protein content) were within a 2kb genomic region or a single haplotype block, it seems that this region represents a major QTL associated with protein in cowpea. Probably it is good enough to address or focus on the peak marker only, i.e., Vu3839577. In addition, more discussion may be added on its effects and potential use.

3.      Additional editing is needed to improve the language and expressions throughout the text. Some examples are listed below, but not limited to them.

4.      Line 61-64, two methods resulted in significant differences (8.5%) in the protein content of the sample set of samples. An explanation can be added to address it.

5.      Line 78, change “to identify a major and six minor QTLs” to “and identified a major and six minor QTLs”.

6.      Line 114, delete “and GS”.

7.      Line 119-120, re-write the sentence because the RCBD is not used to determine the trait, protein or nitrogen content, but for the field experiment implementation.

8.      Line 123, 24 accession should be 24 accessions.

9.      Line 128, what does “genotype x environment heritability” mean? My understanding is that broad-sense heritability based on genotype was estimated in this study.

1.    Line 174, in (what?) is up to.

1.   Line 246-247, how did you calculate the heritability without considering the genotype x environment interaction for your two-location trials? What is the advantage for each case?

1.   Line 254, change “However” to “Moreover”.

1.   Section 4.1. provide the year when the experiments were conducted.

1.   Line 351-352, may change to ….. from 161 cowpea accessions grown at two locations each with three replications as described above.

1.   4.5. section, the heading should not be Phenotyping but GWAS and candidate gene identification.  

1Additional editing is needed to improve the language and expressions throughout the text. Some examples are listed below, but not limited to them.

2.      Line 61-64, two methods resulted in significant differences (8.5%) in the protein content of the sample set of samples. An explanation can be added to address it.

3.      Line 78, change “to identify a major and six minor QTLs” to “and identified a major and six minor QTLs”.

4.      Line 114, delete “and GS”.

5.      Line 119-120, re-write the sentence because the RCBD is not used to determine the trait, protein or nitrogen content, but for the field experiment implementation.

6.      Line 123, 24 accession should be 24 accessions.

7.      Line 128, what does “genotype x environment heritability” mean? My understanding is that broad-sense heritability based on genotype was estimated in this study.

8.      Line 174, in (what?) is up to.

9.      Line 246-247, how did you calculate the heritability without considering the genotype x environment interaction for your two-location trials? What is the advantage for each case?

1.   Line 254, change “However” to “Moreover”.

1.   Section 4.1. provide the year when the experiments were conducted.

1.   Line 351-352, may change to ….. from 161 cowpea accessions grown at two locations each with three replications as described above.

1.   4.5. section, the heading should not be Phenotyping but GWAS and candidate gene identification.  

Author Response

Reply to the reviewer 1

  1. The population size seems a little bit small, consisting of 161 germplasm accessions, especially compared with other major crops like corn and wheat. However, it should be OK as cowpea is a minor crop and has not been investigated so extensively, in particular its germplasm pool is limited. The authors may discuss more about the diversity and the genetic variability of this population.

Reply: Thank you for the kindness reminding. We added more details to this section at L221-226.

  1. Because all the seven SNP markers identified to be associated with the trait (protein content) were within a 2kb genomic region or a single haplotype block, it seems that this region represents a major QTL associated with protein in cowpea. Probably it is good enough to address or focus on the peak marker only, i.e., Vu3839577. In addition, more discussion may be added on its effects and potential use.

Reply: Thank you for the kindness reminding. We added more details to this section at L343-349.

  1. Additional editing is needed to improve the language and expressions throughout the text. Some examples are listed below, but not limited to them.
  2. Line 61-64, two methods resulted in significant differences (8.5%) in the protein content of the sample set of samples. An explanation can be added to address it.

Reply: Appreciate for the thoughtful advice. We rewrite this part at L61-62.

  1. Line 78, change “to identify a major and six minor QTLs” to “and identified a major and six minor QTLs”.

Reply: Thank you for your suggestion. We modified this one.

  1. Line 114, delete “and GS”.

Reply: Thank you. We deleted this one.

  1. Line 119-120, re-write the sentence because the RCBD is not used to determine the trait, protein or nitrogen content, but for the field experiment implementation.

Reply: Thank you so much. We re-wrote this sentence.

  1. Line 123, 24 accession should be 24 accessions.

Reply: Thank you. We fixed this issue.

  1. Line 128, what does “genotype x environment heritability” mean? My understanding is that broad-sense heritability based on genotype was estimated in this study.

Reply: Thank you so much for your professional advice. We fixed this issue.

  1. Line 174, in (what?) is up to.

Reply: Appreciate! We fixed this issue.

11. Line 246-247, how did you calculate the heritability without considering the genotype x environment interaction for your two-location trials? What is the advantage for each case?

Reply: Thank you for your professional advice. We fixed this issue and re-wrote this sentence.

  1. Line 174, in (what?) is up to.

Thank you. We fixed it.

12. Line 254, change “However” to “Moreover”.

Reply: Modified as your suggestion.

  1. Section 4.1. provide the year when the experiments were conducted.

Thank you. The year was added.

14 . Line 351-352, may change to ….. from 161 cowpea accessions grown at two locations each with three replications as described above.

Thank you. The year was added.

  1. 4.5. section, the heading should not be Phenotyping but GWAS and candidate gene identification.

Thank you so much! We fixed this issue.

Reviewer 2 Report

Minor revision

Title:

A Genome-Wide Association Study Reveals Regions Associated with Seed Protein Content in Cowpea

Abstract:

L19: Walp.,

L20: and serves as fodder

L21: diet; therefore,

L23: on 161 cowp

L26: A total of seven

Please modify conclusion in the end of abstract

Introduction:

L53-54: Adding cowpea to the children’s diet can significantly reduce the developmental diseases such as malnutrition and infection in developing areas (Add reference).

L54-56: Its protein can also be used in various food products, including meat alternatives (Add reference), baked goods (Add reference), and protein bars (Add reference), creating economic opportunities for food companies and entrepreneurs.

L59-60: Ddamulira et al. [7] measured the protein content of 30 cowpea genotypes and found that the average cowpea protein content was between 23.9% and 30.3%.

L60: Weng et al (Modified it according to author instruction) and missing in list of references.

L64-65: Boukar et al. [9] measured the protein content of 1541 germplasm lines, and the results were 17.5-32.5%, with an average of 25.0% and a standard deviation of 0.4g. Please modify all references according to the author’s instruction.

Results

Figure 2, 3, 4 replace it with another high resolution

Discussion

L 229: According to previous research, cowpea seeds typically contain 25% protein (Add reference).

L 239: Please modified it according to the author's instructions as mentioned in the introduction section

Materials and Methods

L 350: Please provide details for Seed protein content assessment

References

L469: Modified it according to author instruction.

Author Response

Reply to reviewer 2

Title:A Genome-Wide Association Study Reveals Regions Associated with Seed Protein Content in Cowpea

Abstract:

L19: Walp.,

L20: and serves as fodder

L21: diet; therefore,

L23: on 161 cowp

L26: A total of seven

Please modify conclusion in the end of abstract

Reply: Thank you for the kindness modifications. All the issues were fixed as you suggested.

Introduction:

L53-54: Adding cowpea to the children’s diet can significantly reduce the developmental diseases such as malnutrition and infection in developing areas (Add reference).

L54-56: Its protein can also be used in various food products, including meat alternatives (Add reference), baked goods (Add reference), and protein bars (Add reference), creating economic opportunities for food companies and entrepreneurs.

L59-60: Ddamulira et al. [7] measured the protein content of 30 cowpea genotypes and found that the average cowpea protein content was between 23.9% and 30.3%.

L60: Weng et al (Modified it according to author instruction) and missing in list of references.

L64-65: Boukar et al. [9] measured the protein content of 1541 germplasm lines, and the results were 17.5-32.5%, with an average of 25.0% and a standard deviation of 0.4g. Please modify all references according to the author’s instruction.

Reply: Thank you. All the issues were fixed as the author’s instruction.

Results

Figure 2, 3, 4 replace it with another high resolution

Reply: All the high resolution figures were uploaded.

Discussion

L 229: According to previous research, cowpea seeds typically contain 25% protein (Add reference).

L 239: Please modified it according to the author's instructions as mentioned in the introduction section

Reply: Thank you. All the citations were modified as the author’s instruction.

Materials and Methods

L 350: Please provide details for Seed protein content assessment

Reply: Thank you for the kindness reminding. We added more details to this section at L372-376.

References

L469: Modified it according to author instruction.

Reply: Thank you. All the references were modified as the author’s instruction.
